# Hearing and cognitive decline in aging differentially impact neural tracking of context-supported versus random speech across linguistic timescales

Elena Bolt[1,2]*, Katarina Kliestenec[1], Nathalie Giroud[1,2,3]

**1** Computational Neuroscience of Speech and Hearing, Department of Computational Linguistics, University of Zurich, Zurich, Switzerland, **2** International Max Planck Research School on the Life Course (IMPRS LIFE), University of Zurich, Zurich, Switzerland, **3** Language & Medicine Centre Zurich, Competence Centre of Medical Faculty and Faculty of Arts and Sciences, University of Zurich, Zurich, Switzerland

\* elena.bolt.uzh@gmail.com

**Data Availability Statement:** Data and code availability: The stimulus materials, their transcripts, trigger time points, preprocessed EEG data, anonymized participant data, and data frames

## Abstract

Cognitive decline and hearing loss are common in older adults and often co-occur while investigated separately, affecting the neural processing of speech. This study investigated the interaction between cognitive decline, hearing loss, and contextual cues in speech processing. Participants aged 60 years and older were assessed for cognitive decline using the Montreal Cognitive Assessment and for hearing ability using a four-frequency pure tone average. They listened to in-house–designed matrix-style sentences that either provided supportive context or were random, while we recorded their electroencephalography. Neurophysiological responses were analyzed through auditory evoked potentials and speech tracking at different linguistic timescales (i.e., phrase, word, syllable and phoneme rate) using phase-locking values. The results showed that cognitive decline was associated with decreased response accuracy in a speech recognition task. Cognitive decline significantly impacted the P2 component of auditory evoked potentials, while hearing loss influenced speech tracking at the word and phoneme rates, but not at the phrase or syllable rates. Contextual cues enhanced speech tracking at the syllable rate. These findings suggest that cognitive decline and hearing loss differentially affect the neural mechanisms underlying speech processing, with contextual cues playing a significant role in enhancing syllable rate tracking. This study emphasises the importance of considering both cognitive and auditory factors when studying speech processing in older people and highlights the need for further research to investigate the interplay between cognitive decline, hearing loss and contextual cues in speech processing.

## Introduction

Age-related cognitive decline and hearing loss are prevalent among older adults, with their occurrence increasing alongside global life expectancy [1, 2]. A well-documented link exists

used for statistical analyses are available in the Open Science Framework repository at \href {https://osf.io/5usgp/}{OSF.io/5usgp}. Python and R codes for data processing, neurophysiological analyses, and statistical analyses are published at \href{https://github.com/elbolt/neuroscales}{github.com/elbolt/neuroscales}. Raw EEG data are available upon request from the corresponding author.

**Funding:** This study was funded by the Swiss National Science Foundation, SNSF, www.snf.ch, grant number PR00P1 185715 to Nathalie Giroud.

**Competing interests:** The authors have declared that no competing interests exist.

between cognitive decline and hearing loss [3–5], making hearing loss the most prevalent modifiable risk factor for dementia [6]. At the same time, many older individuals with cognitive impairment also experience hearing loss [7, 8]. Both hearing loss [9–13] and cognitive decline [14–18] significantly impact how the aging brain processes spoken language. Despite this, few studies have used neurophysiological measures to investigate how the older brain processes speech in the presence of both, cognitive decline and hearing loss. Nevertheless, the body of research on this topic is gradually expanding (e.g., [18–22]).

Early signs of cognitive decline might be associated with altered encoding of auditory input, such as repetitive syllable sounds, at both lower-order subcortical stages of processing (e.g., the auditory brainstem and midbrain) and higher-order cortical stages of processing (e.g., the auditory cortex) [18]. To build on these findings, we conducted a follow-up study using natural continuous speech instead of syllables, aiming to elicit subcortical and cortical auditory responses. However, the follow-up study did not confirm altered encoding in cognitive decline [21].

It remains unclear whether auditory evoked components (AEPs) and their P1–N1–P2 complex—distinctive peaks in the AEP waveform used to assess auditory cortex function [23]—can serve as reliable markers of cognitive decline. Morrison et al. reported that the N1 and P2 components did not consistently differ between patients with mild cognitive impairment (MCI), Alzheimer's disease (AD), and healthy controls [20]. In contrast, AEP-based P1, mismatch negativity-related N200, and P300 components derived from active and passive oddball paradigms (which present rare deviant stimuli in a sequence of frequent standard stimuli) may hold more promise as biomarkers of cognitive decline. Additionally, the N400 component, an indicator of semantic processing, may provide further insights into the effects of cognitive decline on speech processing [24], as its abnormalities have been linked to an increased risk of conversion from MCI to AD [19].

It is also well established that enhanced contextual cues facilitate speech perception in older adults [25]. Contextual cues can be linguistic (e.g., sentence context) or non-linguistic (e.g., acoustic or visual cues) and help older adults with cognitive decline to better understand speech in noisy environments and to predict sentence outcomes [26]. Chauvin et al. found that both individuals with MCI and those with AD benefit from a supportive sentence context (and audiovisual speech cues) when perceiving speech in a noisy environment [27]. While all groups in the study benefited from context and audiovisual cues, those with MCI showed a smaller audiovisual benefit in low-context conditions, and those with AD showed a smaller benefit in high-context conditions compared to healthy older adults, suggesting an interaction of multi-sensory and cognitive processing in adverse listening situations.

In summary, existing neurophysiological studies suggest that while early cognitive decline and hearing loss both disrupt neural processing of speech, further research is needed to identify the potential of neurophysiology-based speech processing markers for detecting cognitive decline. Meanwhile, behavioral studies indicate that older adults with cognitive decline benefit from contextual cues in speech.

Building on these findings, the present study aims to explore how cognitive decline and hearing loss interact to affect the neural processing of speech, specifically focusing on the role of contextual cues. Neural processing of speech involves the brain's ability to track the dynamic properties of the speech signal, which is essential for segmenting continuous speech into discrete chunks of linguistic units [28, 29]. The concept of neural tracking of speech, or "speech tracking," refers to this ability, which can be captured using computational approaches such as connectivity measures [30].

In this study, 45 participants aged 60 to 83 years listened to two sets of in-house–designed, matrix-style sentences (i.e., sentences with a fixed grammatical structure) that either provided

supportive sentence context or were random, while we recorded their electroencephalography (EEG). The sentences were designed to be more naturalistic than traditional speech stimuli, allowing for the investigation of speech tracking in a more ecologically valid setting, while still following a fixed grammatical structure. Context sentences were designed to make the outcome of the sentence predictable by leading to a specific target noun in the penultimate part of the sentence, whereas the clauses in the random sentences were unrelated. This design enabled us to examine how context influences the processing of speech in older adults with varying degrees of cognitive decline and hearing loss. We assessed early signs of cognitive decline using the Montreal Cognitive Assessment (MoCA) and its clinical cutoff for MCI [18, 31], assigning participants to either a "normal" or "low" MoCA group. Hearing ability was accounted for by measuring the four-frequency pure tone average (PTA) [32].

We analyzed the neurophysiological data in two primary ways. First, we examined the AEPs elicited by the speech stimuli, focusing on the P1–N1–P2 complex related to the onset of the speech signal and the N400-like component related to semantic processing, locked to the target noun in the context vs. random sentences. The N400 component is typically elicited by semantic violations in sentence contexts and reflects the brain's response to unexpected or incongruent words [24]. Investigating a N400-like component allowed us to assess whether our speech stimuli could elicit a similar response, indicating the processing of semantic information. Additionally, we analyzed the neural tracking of speech at different linguistic timescales, following the approach by Keitel et al. [33]. Specifically, we focused on the neural tracking of speech at the phrase, word, syllable, and phoneme rates presented in the matrix sentences (see Fig 1A). This multi-level analysis of neural tracking provides a comprehensive understanding of how cognitive decline and hearing loss affect the brain's ability to follow the dynamic properties of the speech signal, encompassing various levels of linguistic complexity.

Our main goal was to explore how cognitive status and hearing ability in older adults affect the neural tracking of context and random speech. By analyzing speech tracking across various linguistic timescales, we aimed to provide a deeper understanding of how cognitive decline and hearing loss influence speech processing in naturalistic listening environments. We hypothesized that participants with early signs of cognitive decline would show altered tracking of speech, as evidenced by differences in AEPs and neural tracking of speech at different

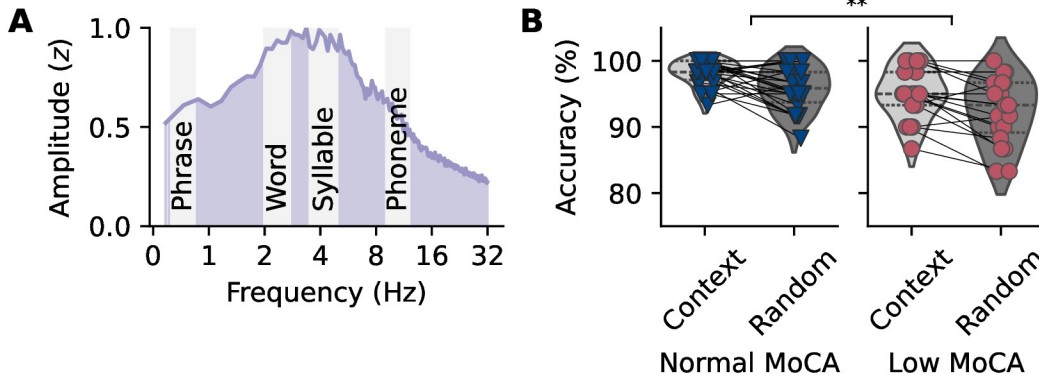

**Fig 1. Speech material modulation spectrum and behavioral speech recognition task results. A** Temporal modulation spectrum of all speech stimuli, with the ranges for the phrase, word, syllable, and phoneme rates highlighted in gray. The modulation spectrum shows increased energy at lower modulation rates, corresponding to the slower rates of the speech material. **B** Correct response rates for the Montreal Cognitive Assessment (MoCA) groups in the context and random conditions. Participants with signs of cognitive decline (low MoCA group) were less likely to answer correctly compared to participants in the normal MoCA group, independent of contextual cues in the speech material.

linguistic timescales compared to participants without cognitive decline. We also anticipated that hearing ability would modulate speech tracking, with participants with poorer hearing showing reduced speech tracking compared to those with better hearing. Finally, we expected that the presence of contextual cues would enhance speech tracking in participants with early signs of cognitive decline and hearing loss, as these cues have been shown to facilitate speech perception in older adults with cognitive decline [27].

## Results

### Behavioral performance in speech recognition task affected by cognitive decline

After each matrix sentence, participants were asked to identify a target word from a list of four words. The main goal of this behavioral speech recognition task after each stimulus was to maintain their attention and ensure that they listened. Participants performed well, answering correctly in 96±4% of the trials. No participant scored below 85% correct responses. The task was designed to be easy, and we excluded all trials with incorrect responses from neurophysiological analyses. We analyzed the effects of MoCA group, PTA, and condition on the correct response at the item level using a generalized linear mixed model (GLMM) that accounted for the nested structure of the data, which included correct responses measured across different participants and conditions. The results of the GLMM are summarized in Table 1.

We found a significant main effect of MoCA group, with the low MoCA group having lower odds of responding correctly compared to the normal MoCA group (Fig 1B). No significant main effects were found for PTA, condition, or the control variable age, nor was the interaction between MoCA group and PTA significant. Participants in the normal MoCA group reached an average correct response rate of 98±2% in the context condition and 96±3% in the random condition, while participants in the low MoCA group reached an average correct response rate of 95±4% in the context condition and 92±5% in the random condition.

Thus, the results suggest that participants with signs of cognitive decline were less likely to answer correctly, while hearing ability did not affect their performance. The supporting context also had no influence on how well the participants could follow the sentence, i.e., they could understand the speech just as well without additional context.

### Evoked potentials in relation to cognitive decline and hearing ability

**P1, N1, and P2 components elicited by speech onset.**  We identified the average P1 component at 57±10 ms, the N1 component at 119±18 ms, and the P2 component at 247±29 ms after stimulus onset. Grand average responses to speech onset are shown in Fig 2A, with

**Table 1. Results of the generalized linear mixed model (GLMM) for correct responses in the speech recognition task.** Responses were coded as 0 (incorrect) or 1 (correct). OR, odds ratio; CI, confidence interval; LL, lower limit; UL, upper limit; MoCA, Montreal Cognitive Assessment; PTA, pure-tone average. Significance levels: $p < 0.001$, ***; $p < 0.01$, **.

|  | 95% CI | | | z | p |  |
|---|---|---|---|---|---|---|
|  | *OR* | *LL* | *UL* |  |  |  |
| Intercept | 143.45 | 73.42 | 280.29 | 14.531 | <0.001 | *** |
| MoCA group (low) | 0.44 | 0.27 | 0.73 | −3.209 | 0.001 | ** |
| PTA ($z$) | 0.74 | 0.52 | 1.03 | −1.768 | 0.077 |  |
| Age ($z$) | 0.61 | 0.30 | 1.25 | −1.359 | 0.174 |  |
| Condition (random) | 0.90 | 0.67 | 1.20 | −0.714 | 0.475 |  |
| MoCA group × PTA | 1.18 | 0.73 | 1.90 | 0.675 | 0.500 |  |

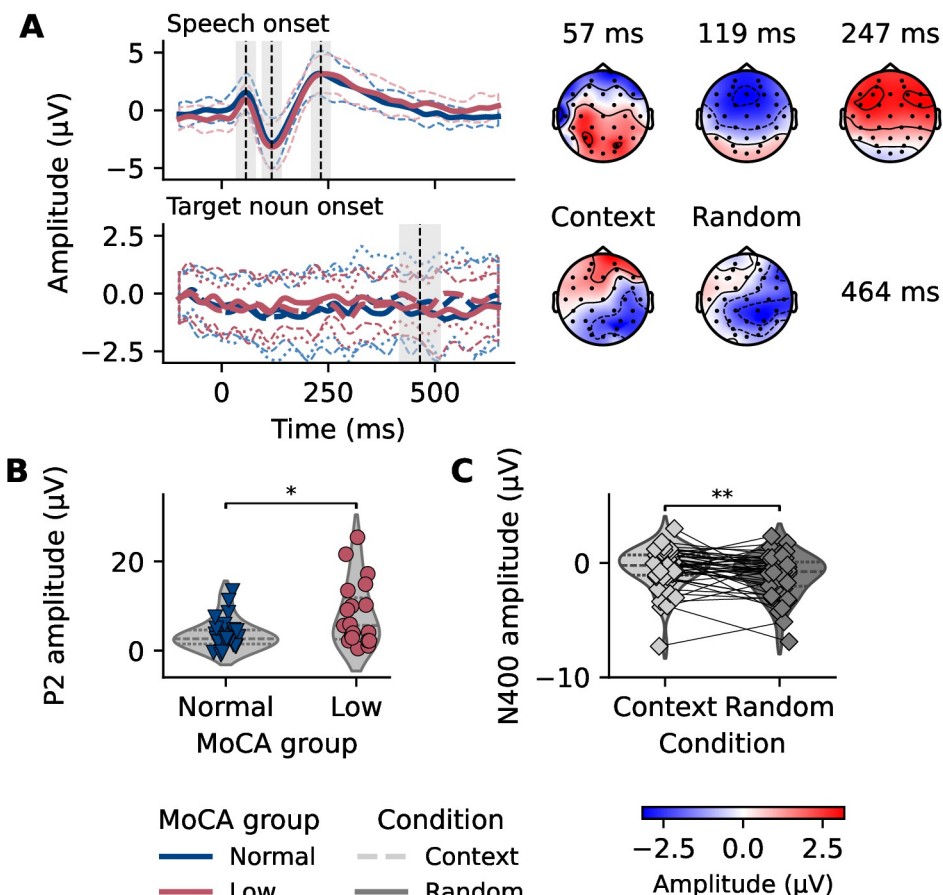

**Fig 2. Responses to speech onset and target nouns in relation to cognitive decline and context. A** Grand average evoked potentials time-locked to speech onset (upper panel) and to the target noun (lower panel), color-coded by Montreal Cognitive Assessment (MoCA) group. Speech onset responses were averaged across all trials, while target noun responses were averaged across trials within the context and random conditions. Dashed vertical lines indicate the P1, N1, P2, and N400-like components, with shaded areas highlighting the time windows represented in the topographical maps for each component (50 ms for P1–N1–P2 and 100 ms for N400). Responses were filtered between 1–12 Hz (speech onset responses) and 0.1–20 Hz (target noun responses) for visualization purposes only. Speech onset responses were averaged across an auditory frontotemporal cluster consisting of electrodes F3, FC1, FC5, FC6, FC2, and F4. Target noun responses were averaged across a centroparietal cluster of electrodes Cz, CP1, CP2, Pz, and P3. Topographical maps represent the mean amplitude at the grand average peak for each component, independent of MoCA group. **B** P2 amplitudes were significantly larger in the low MoCA group compared to the normal MoCA group. **C** The N400-like response was significantly more negative in the random condition compared to the context condition.

topographical maps of the P1, N1, and P2 components. All components were unambiguously identifiable for all participants. Latencies did not significantly differ between MoCA groups, as indicated by two-tailed $t$-tests (P1: $t(43) = -0.6$, $p = 0.557$, $d = -0.2$, normal MoCA group: 57 ms, low MoCA group: 56 ms; N1: $t(43) = -1.4$, $p = 0.168$, $d = -0.4$, normal MoCA group: 122 ms, low MoCA group: 114 ms; P2: $t(43) = 2.0$, $p = 0.055$, $d = 0.6$, normal MoCA group: 240 ms, low MoCA group: 257 ms). Linear regression models for each peak amplitude are summarized in Table 2. Notably, the MoCA group had a significant effect on the P2 component, but not on the P1 or N1 components, suggesting that individuals with signs of cognitive decline exhibit enlarged P2 responses (Fig 2B). Hearing ability, quantified by PTA, did not significantly affect the P1, N1, or P2 amplitudes. Additionally, older age was associated with increased P1, N1, and P2 amplitudes.

**Table 2. Results of the linear models for the amplitudes (in $\mu$V) of the components P1, N1, P2, and N400.** CI, confidence interval; LL, lower limit; UL, upper limit; MoCA, Montreal Cognitive Assessment; PTA, four-frequency pure-tone average. Significance levels: $p < 0.001$, ***; $p < 0.01$, **; $p < 0.05$, *.

| | *Estimate* | 95% CI | | *t* | *p* | |
| --- | --- | --- | --- | --- | --- | --- |
| | | *LL* | *UL* | | | |
| **P1** | | | | | | |
| Intercept | 1.629 | 0.059 | 3.198 | 2.1 | 0.042 | * |
| MoCA group (low) | 1.918 | −0.542 | 4.377 | 1.6 | 0.123 | |
| PTA (z) | 0.313 | −1.327 | 1.953 | 0.4 | 0.702 | |
| Age (z) | 2.540 | 1.100 | 3.981 | 3.6 | 0.001 | ** |
| MoCA group × PTA | −0.809 | −3.263 | 1.645 | −0.7 | 0.509 | |
| **N1** | | | | | | |
| Intercept | −0.431 | −2.047 | 1.185 | −0.5 | 0.593 | |
| MoCA group (low) | 2.519 | −0.013 | 5.050 | 2.0 | 0.051 | |
| PTA (z) | 0.006 | −1.682 | 1.695 | 0.0 | 0.994 | |
| Age (z) | 1.781 | 0.298 | 3.263 | 2.4 | 0.020 | * |
| MoCA group × PTA | −0.670 | −3.196 | 1.857 | −0.5 | 0.595 | |
| **P2** | | | | | | |
| Intercept | 4.002 | 2.015 | 5.988 | 4.1 | <0.001 | *** |
| MoCA group (low) | 3.586 | 0.473 | 6.698 | 2.3 | 0.025 | * |
| PTA (z) | 1.032 | −1.043 | 3.108 | 1.0 | 0.321 | |
| Age (z) | 2.266 | 0.443 | 4.089 | 2.5 | 0.016 | * |
| MoCA group × PTA | −1.130 | −4.236 | 1.977 | −0.7 | 0.467 | |
| **N400** | | | | | | |
| Intercept | 0.669 | −0.059 | 1.398 | 1.9 | 0.071 | |
| MoCA group (low) | −0.047 | −1.189 | 1.094 | −0.1 | 0.934 | |
| PTA (z) | −0.505 | −1.262 | 0.252 | −1.3 | 0.185 | |
| Age (z) | 0.139 | −0.526 | 0.803 | 0.4 | 0.675 | |
| MoCA group × PTA | 0.701 | −0.431 | 1.834 | 1.3 | 0.218 | |

**N400-like responses elicited by context vs. random condition.** We observed a component resembling the N400 at 464±98 ms after target noun onset. Fig 2A shows the difference wave of the grand average responses locked to the target nouns in the context and random conditions and the topographical distribution of the N400-like response. Latencies did not significantly differ between MoCA groups (two-tailed *t*-test, $t(43) = 0.6$, $p = 0.521$, $d = 0.2$, normal MoCA group: 456 ms, low MoCA group: 476 ms). To evaluate the N400-like response, we compared the mean amplitudes of the context and random conditions. In the random condition, the mean amplitude was significantly more negative compared to the context condition (one-tailed paired *t*-test, $t(44) = -2.5$, $p = 0.008$, $d = -0.4$), as shown in Fig 2C. This confirmed that the random speech material elicited a component resembling the N400 response to the target nouns. The results of a linear model on the N400-like negativity are summarized in Table 2. Neither MoCA group, PTA, age, nor their interactions had significant effects on the N400-like amplitude, suggesting that none of these factors influenced the N400-like response in this study.

## Speech tracking across linguistic timescales as a function of cognitive decline and hearing ability

To analyze the neural tracking of speech across different linguistic timescales, we calculated the phase-locking value (PLV) for the phrase, word, syllable, and phoneme rates. We

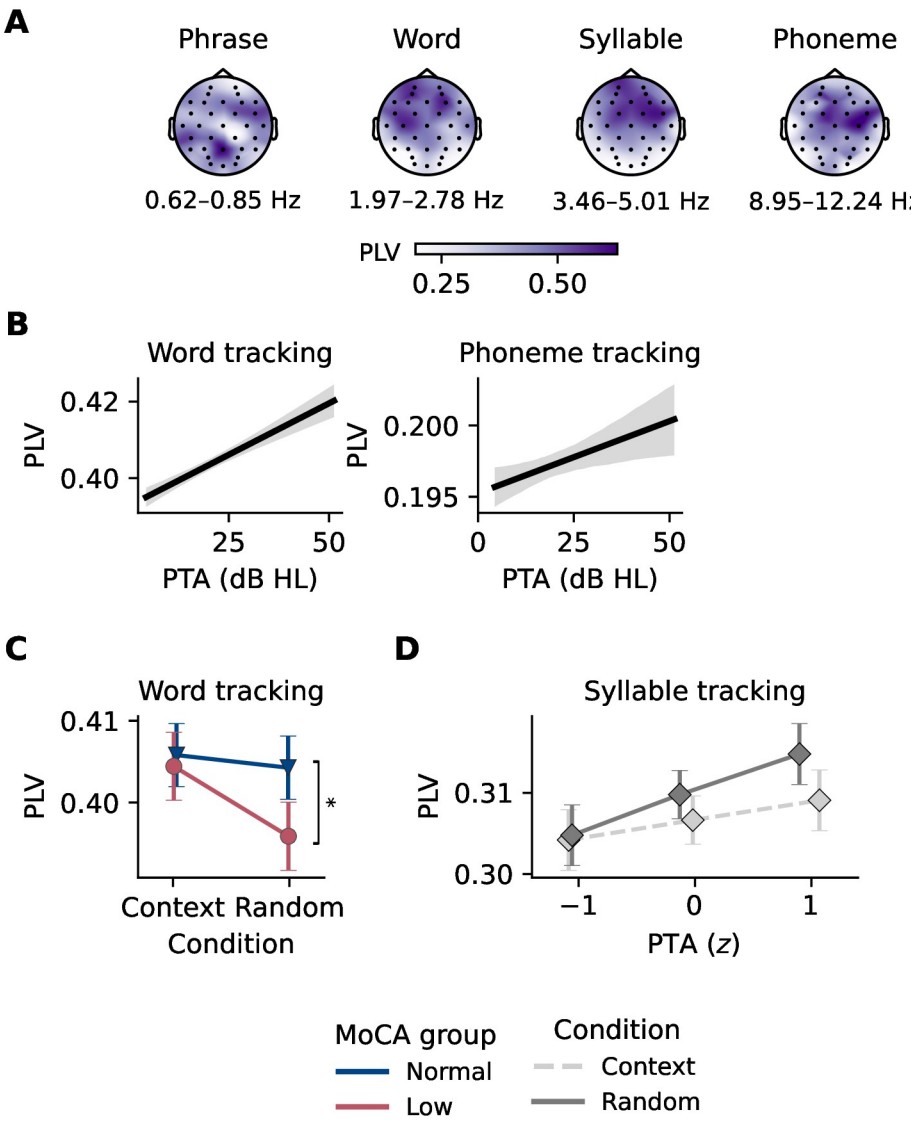

**Fig 3. Neural tracking of speech across linguistic timescales in relation to cognitive decline and hearing ability. A** Topographical distribution of the phase-locking values (PLV) for phrase, word, syllable, and phoneme rates across all trials and conditions. **B** Relationship between four-frequency pure tone average (PTA) and word rate tracking (left) and phoneme rate tracking (right), showing a significant effect of hearing ability on both rates. The shaded area represents the 95% confidence interval. **C** Interaction between Montreal Cognitive Assessment (MoCA) group and condition for word rate tracking, showing a significant difference in the random condition between the normal and low MoCA groups. Speech tracking was significantly higher in the normal MoCA group compared to the low MoCA group. **D** Interaction between PTA and condition for syllable rate tracking, illustrating the effects of hearing ability and contextual cues. In the context condition, syllable rate was significantly higher compared to the random condition. The effect of PTA on syllable rate tracking was, however, not significant according to post-hoc tests.

calculated the PLV between the speech envelope and the EEG signal for both conditions separately. The four rates and their topographical distribution across all trials (independent of condition) are shown in Fig 3A. The topographical maps indicate that the tracking values were most pronounced in frontocentral electrodes, though with a less distinct distribution than seen in the speech onset responses. In the statistical analyses, we included three electrode clusters, a frontal (F), central (C), and parietal (P) cluster, to account for the topographical

distribution of the tracking values. We analyzed the effects of MoCA group, PTA, and condition on speech tracking using linear mixed models (LMMs) that accounted for the nested data structure, including speech tracking values measured across electrode clusters and conditions. This model also accounted for random effects for participants and trials. The results of the fixed effects of the LMMs for each linguistic timescale-specific speech tracking value are summarized in Table 3.

**Speech tracking at the phrase rate.** Our model on phrase rate tracking revealed no significant main effects of MoCA group, PTA, or condition. Only the control variable age had a significant effect on phrase rate tracking, with older age being associated with a significant decrease in tracking at the phrase rate. Thus, phrase rate tracking was not directly affected by cognitive decline or hearing ability, nor was it facilitated by increased context in the speech stimuli.

**Speech tracking at the word rate.** The model on word rate tracking revealed a significant main effect of PTA (Fig 3B), but not of MoCA group, suggesting that hearing ability significantly affected word rate tracking, while cognitive decline did not. We found a main effect for the parietal cluster, which showed a significant decrease in tracking at the word rate, indicating that speech tracking was mainly driven by frontocentral electrodes (Fig 3A). Notably, there was a significant interaction between MoCA group and condition, suggesting that the effect of condition on word rate tracking differed between MoCA groups. Post-hoc tests using estimated marginal means (EMMs) revealed that within the context condition, there was no significant difference in word rate tracking between the normal and low MoCA groups ($EMM = 0.001$, $p = 0.722$). However, in the random condition, word rate tracking was significantly higher in the normal MoCA group compared to the low MoCA group ($EMM = 0.008$, $p = 0.030$, Fig 3C).

While word rate tracking was not directly affected by cognitive decline, it increased with increased hearing impairment Moreover, the effect of context on word rate tracking was more pronounced in individuals with signs of cognitive decline, suggesting that context facilitated word rate tracking in participants with cognitive decline.

**Speech tracking at the syllable rate.** We observed a significant main effect of condition on syllable rate tracking, with the random condition leading to increased tracking at the syllable rate. The parietal cluster showed a significant decrease in tracking at the syllable rate, again indicating that this speech tracking was mainly driven by frontocentral electrodes (Fig 3A). However, there were no main effects of MoCA group or PTA on syllable rate tracking. In line with word rate-based speech tracking results, there was a significant interaction between MoCA group and condition. Post-hoc tests revealed no significant difference in syllable rate tracking between the normal and low MoCA groups within the context condition ($EMM = −0.006$, $p = 0.138$) or within the random condition ($EMM = 0.006$, $p = 0.135$).

Furthermore, we found a significant interaction between PTA and condition, shown in Fig 3D. Post-hoc tests revealed that within the context condition, syllable rate tracking did not significantly differ across PTA levels (PTA −1 vs. 0: $EMM = −0.002$, $p = 0.527$; PTA −1 vs. 1: $EMM = −0.005$, $p = 0.527$; PTA 0 vs. 1: $EMM = −0.002$, $p = 0.527$). Similarly, within the random condition, syllable rate tracking did not significantly differ across PTA levels (PTA −1 vs. 0: $EMM = 0.003$, $p = 0.072$; PTA −1 vs. 1: $EMM = 0.006$, $p = 0.072$; PTA 0 vs. 1: $EMM = 0.003$, $p = 0.072$), suggesting that hearing ability did not affect tracking at the syllable rate in the context condition.

Overall, these results suggest that tracking of the syllable rate was not affected by cognitive decline or hearing ability but was facilitated by increased context in the speech stimuli.

**Speech tracking at the phoneme rate.** In line with the results for the word rate, we found a significant main effect of PTA on phoneme rate tracking (Fig 3B), but not of MoCA group,

**Table 3. Results of linear mixed models (LMMs) on linguistic timescale tracking for the phrase, word, syllable, and phoneme rates.** CI, confidence interval; LL, lower limit; UL, upper limit; MoCA, Montreal Cognitive Assessment; PTA, four-frequency pure-tone average; C, central, P, parietal. Significance levels: $p < 0.001$, ***; $p < 0.01$, **; $p < 0.05$, *.

| | *Estimate* | 95% CI | | *df* | *t* | *p* | |
| --- | --- | --- | --- | --- | --- | --- | --- |
| | | *LL* | *UL* | | | | |
| **Phrase rate at 0.62–0.85 Hz** | | | | | | | |
| Intercept | 0.625 | 0.596 | 0.653 | 135.1 | 42.4 | <0.001 | *** |
| MoCA group (low) | −0.001 | −0.013 | 0.012 | 52.3 | −0.1 | 0.954 | |
| PTA (z) | −0.001 | −0.009 | 0.007 | 51.3 | −0.2 | 0.869 | |
| Condition (random) | −0.003 | −0.043 | 0.036 | 120 | −0.2 | 0.876 | |
| Cluster (C) | −0.003 | −0.009 | 0.002 | 46279.8 | −1.3 | 0.209 | |
| Cluster (P) | −0.001 | −0.005 | 0.005 | 46279.8 | −0.0 | 0.963 | |
| Age (z) | −0.008 | −0.014 | −0.001 | 39.9 | −2.3 | 0.029 | * |
| MoCA group × PTA | 0.008 | −0.004 | 0.020 | 52.6 | 1.3 | 0.214 | |
| MoCA group × Condition | 0.003 | −0.006 | 0.011 | 46285.3 | 0.6 | 0.570 | |
| PTA × Condition | −0.002 | −0.008 | 0.003 | 46287.5 | −0.8 | 0.406 | |
| MoCA group × PTA × Condition | 0.004 | −0.005 | 0.013 | 46286.2 | 0.9 | 0.368 | |
| **Word rate at 1.97–2.78 Hz** | | | | | | | |
| Intercept | 0.408 | 0.400 | 0.416 | 171.5 | 102.1 | <0.001 | *** |
| MoCA group (low) | −0.001 | −0.009 | 0.006 | 55.9 | −0.4 | 0.723 | |
| PTA (z) | 0.007 | 0.002 | 0.012 | 54.6 | 2.7 | 0.010 | * |
| Condition (random) | −0.002 | −0.011 | 0.008 | 138 | −0.3 | 0.741 | |
| Cluster (C) | −0.002 | −0.006 | 0.001 | 46280.1 | −1.1 | 0.251 | |
| Cluster (P) | −0.005 | −0.009 | −0.001 | 46280.1 | −2.7 | 0.007 | ** |
| Age (z) | −0.002 | −0.006 | 0.002 | 39.9 | −0.9 | 0.372 | |
| MoCA group × PTA | 0.004 | −0.003 | 0.012 | 56.4 | 1.1 | 0.279 | |
| MoCA group × Condition | −0.007 | −0.013 | −0.001 | 46304.1 | −2.3 | 0.022 | * |
| PTA × Condition | −0.004 | −0.008 | 0.001 | 46315.3 | −1.9 | 0.062 | |
| MoCA group × PTA × Condition | −0.002 | −0.008 | 0.004 | 46309.5 | −0.7 | 0.507 | |
| **Syllable rate at 3.46–5.01 Hz** | | | | | | | |
| Intercept | 0.307 | 0.300 | 0.314 | 126 | 87.7 | <0.001 | *** |
| MoCA group (low) | 0.006 | −0.002 | 0.014 | 47.5 | 1.5 | 0.144 | |
| PTA (z) | 0.002 | −0.004 | 0.007 | 46.9 | 0.6 | 0.546 | |
| Condition (random) | 0.009 | 0.002 | 0.016 | 139.1 | 2.6 | 0.010 | * |
| Cluster (C) | −0.002 | −0.005 | 0.001 | 46280.1 | −1.4 | 0.168 | |
| Cluster (P) | −0.009 | −0.011 | −0.006 | 46280.1 | −6.0 | <0.001 | *** |
| Age (z) | 0.004 | 0.001 | 0.009 | 39.9 | 2.0 | 0.052 | |
| MoCA group × PTA | 0.002 | −0.006 | 0.009 | 47.7 | 0.4 | 0.675 | |
| MoCA group × Condition | −0.012 | −0.017 | −0.007 | 46302.9 | −5.1 | <0.001 | *** |
| PTA × Condition | 0.004 | 0.001 | 0.007 | 46314.1 | 2.8 | 0.005 | ** |
| MoCA group × PTA × Condition | −0.003 | −0.008 | 0.001 | 46308.5 | −1.4 | 0.149 | |
| **Phoneme rate at 8.95–12.24 Hz** | | | | | | | |
| Intercept | 0.197 | 0.193 | 0.200 | 120.3 | 102.4 | <0.001 | *** |
| MoCA group (low) | 0.002 | −0.003 | 0.006 | 52.4 | 0.7 | 0.461 | |
| PTA (z) | 0.003 | 0.001 | 0.006 | 51.4 | 2.1 | 0.037 | * |
| Condition (random) | 0.001 | −0.003 | 0.004 | 158.6 | 0.3 | 0.762 | |
| Cluster (C) | 0.001 | −0.001 | 0.003 | 46279.2 | 1.3 | 0.202 | |
| Cluster (P) | −0.001 | −0.003 | 0.001 | 46279.2 | −0.8 | 0.430 | |
| Age (z) | −0.001 | −0.003 | 0.002 | 40.1 | −0.7 | 0.502 | |
| MoCA group × PTA | −0.001 | −0.005 | 0.004 | 52.8 | −0.2 | 0.804 | |

**Table 3.** (Continued)

| | Estimate | 95% CI | | df | t | p | |
| --- | --- | --- | --- | --- | --- | --- | --- |
| | | LL | UL | | | | |
| MoCA group × Condition | −0.004 | −0.007 | −0.001 | 46319.5 | −2.3 | 0.019 | * |
| PTA × Condition | −0.003 | −0.006 | −0.001 | 46333.3 | −3.2 | 0.001 | ** |
| MoCA group × PTA × Condition | 0.002 | −0.002 | 0.005 | 46327.1 | 0.9 | 0.356 | |

indicating that hearing led to increased tracking at the phoneme rate. Similar to the results for the syllable rate, we found significant interactions between MoCA group and condition, and between PTA and condition. The interaction between MoCA group and condition was not confirmed by post-hoc tests, which showed no significant difference in phoneme rate tracking between the normal and low MoCA groups within the context condition ($EMM = −0.002$, $p = 0.458$) nor within the random condition ($EMM = 0.002$, $p = 0.352$). The same held true for the interaction between PTA and condition, where post-hoc tests showed that within the context condition, phoneme rate tracking did not significantly differ across PTA levels (PTA −1 vs. 0: $EMM = −0.003$, $p = 0.058$; PTA −1 vs. 1: $EMM = −0.006$, $p = 0.058$; PTA 0 vs. 1: $EMM = −0.003$, $p = 0.058$), nor did it differ within the random condition (PTA −1 vs. 0: $EMM = −0.0003$, $p = 0.969$; PTA −1 vs. 1: $EMM = −0.0006$, $p = 0.969$; PTA 0 vs. 1: $EMM = −0.0003$, $p = 0.969$).

Taken together, these results suggest that phoneme rate tracking was not directly affected by cognitive decline but was influenced by hearing ability. As hearing ability decreased, tracking at the phoneme rate increased.

## Discussion

In this study, we investigated how cognitive decline and hearing loss affect the neural processing of speech, specifically comparing conditions with or without contextual cues. We expected differential effects of cognitive decline and hearing loss on speech-evoked neural responses at the early sensory processing level and across linguistic timescales. Behaviorally, our findings indicate that cognitive decline was associated with decreased response accuracy in a behavioral forced-choice speech recognition task, aligning with our hypothesis. Interestingly, the neurophysiological responses provided more variable patterns, with cognitive decline and hearing loss showing distinct effects on speech tracking across different linguistic time scales (i.e., phonemes, syllables, words, sentences).

Specifically, we hypothesized that participants with early signs of cognitive decline would show altered neural processing of speech, as evidenced by differences in AEPs and neural tracking at different linguistic timescales. In the following we discuss in detail which hypotheses were fulfilled and which were not.

### Cognitive decline affected the P2 component in evoked potentials

Our analysis revealed that the P2 amplitudes were significantly larger for participants with signs of early cognitive decline. This finding aligns with Bidelman et al. [18], who reported that the N1–P2 voltage difference is a significant predictor of cognitive decline. However, it contrasts with Morrison et al. [20], who found that the P2 component was not a reliable indicator for MCI or AD. Interestingly, we did not observe an effect of hearing ability on the P2 amplitude or any other component. This is surprising, as previous studies have shown that P2 amplitudes are usually delayed and enlarged in the presence of hearing loss (e.g., [34, 35]).

Notably, the control variable age was significant for all components (P1–N1–P2), indicating that older age was associated with increased amplitudes. Most participants exhibited no to mild hearing impairment, with only seven participants having PTA values greater than 34 dB HL, which is considered moderate hearing loss [36]. Given the age of the participants, their hearing ability might have been relatively preserved, which might explain the lack of significant effects of hearing loss on the AEP components.

Furthermore, our analysis of the N400-like responses did not reveal any effects of cognitive decline or hearing loss. This is unexpected, given the N400's potential to identify patients with MCI at risk for conversion to AD [19], while little is known about N400 changes in early cognitive decline. One potential explanation for the lack of significant findings in the N400-like component could be the mild cognitive decline in the low MoCA group, with nine participants scoring only one point below the clinical cut-off threshold. Furthermore, this finding suggests that further research is needed to fully understand the relationship between cognitive decline, hearing loss, and semantic processing. It is important to note that we refer to this component as N400-*like*, given the specific characteristics of our in-house–designed speech stimuli, which will be further elaborated upon in a following section on the limitations of our study.

## Cognitive decline and hearing loss differentially affected speech tracking

We found that cognitive decline did not directly affect speech tracking at any of the linguistic rates we examined: phrase, word, syllable, and phoneme. This suggests that the neural mechanisms underlying speech tracking remain relatively robust in the presence of early cognitive decline. However, hearing ability did modulate speech tracking strength, specifically at the word and phoneme rates, but not at the phrase or syllable rates. Our findings align with several other studies that have reported an increase in envelope tracking as a measure of speech tracking with decreasing hearing ability [11, 12, 37, 38]. These studies indicate that individuals with hearing loss may exhibit enhanced neural tracking of the speech envelope, potentially as a compensatory mechanism to aid in speech comprehension.

It is important to note that in speech tracking measures such as cross-correlation or temporal response functions, the EEG and envelope signals are often filtered between approximately 1–9 Hz or even more narrowly. This frequency range is critical for understanding speech, which is characterized by pronounced low-frequency envelope modulations (2–8 Hz) [39]. In our study, the phrase rate fell below this frequency range, while the phoneme rate was just within the upper limit. It is thus intriguing that we observed an effect of hearing loss on speech tracking at the phoneme rate but not at the syllable rate, which falls within the critical frequency range for speech comprehension. While this might explain why tracking at the phrasal rate was not affected by hearing loss, it does not fully account for the lack of an effect at the syllable rate. Alternatively, it might imply that tracking at these rates remains relatively unaffected by cognitive decline and hearing loss, suggesting that the neural mechanisms underlying speech tracking at different linguistic timescales are distinct and more robust for syllables as compared to other time scales. Nevertheless, there is one study that incorporated speech tracking at a linguistic level and found that hearing loss led to increased linguistic tracking, specifically at the phoneme level [40], supporting our findings. This suggests that the relationship between hearing loss and speech tracking at different linguistic timescales is complex and warrants further investigation.

## Contextual cues facilitate tracking at the syllable rate and decrease tracking at word rate in cognitive decline

Our study found that contextual cues facilitated speech tracking at the syllable rate, with higher speech tracking values observed in the context condition compared to the random condition.

This effect was not observed at the other linguistic timescales, suggesting a unique interaction between contextual cues and neural processing at the syllable level. The finding that contextual cues enhance syllable rate tracking aligns with the idea that semantic context can influence early auditory encoding of speech. A study by Broderick et al. [41] demonstrated that the early cortical tracking of a word's speech envelope is enhanced by its semantic similarity to its sentential context. This suggests a top-down propagation of information through the processing hierarchy, linking prior knowledge with the early cortical entrainment of words in natural continuous speech. In contrast, the lack of significant effects of contextual cues on speech tracking at the phrase and phoneme rates might suggest that these linguistic timescales rely more on bottom-up processing or may not be as sensitive to top-down influences from contextual information.

Furthermore, we found that the absence of contextual cues led to reduced speech tracking at the word rate in participants with signs of cognitive decline. This suggests that cognitive decline may result in difficulty tracking words in speech when deprived of contextual cues. Given that word recognition in speech demands significant cognitive resources, this task becomes even more demanding in the presence of cognitive decline [42, 43]. These results underscore the critical role of contextual cues in supporting speech comprehension in older adults with cognitive decline.

Chauvin et al. [27] found that individuals with MCI and those with AD benefit from supportive sentence context and audiovisual speech cues when perceiving speech in a noisy environment. While all groups in the study benefited from context and audiovisual cues, those with MCI showed a smaller audiovisual benefit in low-context conditions, and those with AD showed a smaller benefit in high-context conditions compared to healthy older adults. This suggests an interaction of multi-sensory and cognitive processing in adverse listening situations. In contrast, our study observed the facilitative effect of contextual cues only at the syllable rate and not at other linguistic timescales. We expected to see the benefit of contextual cues across all linguistic timescales, given the established role of semantic context in speech processing.

Similarly, Frei et al. [44] demonstrated that neural speech tracking and behavioral comprehension are influenced by the interaction between sensory modality, memory load, and individual auditory working memory capacity in older adults. Their study, using immersive virtual reality to provide natural continuous multisensory speech, showed that under low memory load, neural speech tracking increased in the immersive modality, particularly for individuals with low auditory working memory. These findings underscore the dynamic allocation of sensory and cognitive resources based on the sensory and cognitive demands of speech and individual capacities. This further supports the idea that integrating sensory cues and managing cognitive load is crucial for effective speech processing in older adults.

While these results provide valuable insights, it is important to note that the significance was observed only for a few model terms, indicating that further research is needed to fully understand these interactions.

## Strengths and limitations

**Assessment of cognitive decline through screening tool.** We acknowledge that using the Montreal Cognitive Assessment (MoCA) to assess putative signs of early cognitive decline is a critical and potentially limiting factor in our study. We chose the MoCA because, to date, it is the most widely used quick screening tool for MCI, with an established clinical threshold for identifying patients at risk [31]. Medical practitioners frequently rely on the MoCA to screen their patients, as it is a reliable measure of global cognition in older adults [45]. Moreover, we

aimed to identify early cognitive decline, and the MoCA is particularly suited for detecting subtle cognitive changes at an early stage. However, it is evident that a study including participants with distinct neurocognitive profiles could make much stronger claims. In an ideal scenario, having a healthy control group and patients with clinical diagnoses of MCI and dementia would allow for more robust comparisons. Such a study could also incorporate more comprehensive cognitive testing, including a full-scale IQ test, to corroborate the findings suggested by the MoCA.

**Limited hearing loss gradients reduce generalizability.** Our sample had relatively good hearing for their age, resulting in low variability in hearing loss. While this minimized confounding factors, it may have concealed the effects of hearing loss on speech tracking measures. Consequently, we didn't find consistent significant effects across all measures. Including participants with a broader range of hearing abilities in future studies could provide a clearer understanding of how cognitive decline interacts with hearing loss in speech processing and enhance the generalizability of the findings.

**Constrained naturalness of speech stimuli.** In this study, we attempted to create speech stimuli that reflect somewhat natural speech. We designed matrix-style sentences that, while structured and repetitive, aimed to balance naturalness with experimental control. This fixed structure, although likely transparent to participants after some exposure, allowed us to control for linguistic and semantic content, which was crucial for our PLV analyses. This approach represents an attempt to balance between naturalistic speech and the need for controlled experimental stimuli. However, we recognize that more naturalistic stimuli, such as audiobooks or podcasts, could provide a richer and potentially more engaging listening experience for participants.

**N400-like response may be underrepresented.** We analyzed an N400-like response, which we locked to a target word within our self-designed speech material. This approach allowed us to examine semantic processing in a controlled manner. However, we acknowledge that our speech stimuli were only validated on a behavioral level. The N400-like response observed in Fig 2A, although showing increased negativity approximately 400 ms after the target word onset in the random compared to the context condition, was not as distinctive as seen in other studies with older participants (e.g., [46]). Our study did not employ a traditional N400 paradigm with well-validated stimuli known to elicit robust N400 responses, such as the sentences used in the Speech Perception in Noise Test (SPIN-R) [47]. In SPIN-R, semantic context is manipulated to create high- and low-constraint sentences, with high-constraint sentences providing context that facilitates the processing of the target word [48]. Therefore, we refrain from making claims about the effects of cognitive decline and hearing loss on the N400 response in our study.

**PLV as a measure of speech tracking is limited.** We relied on PLV to quantify speech tracking, which measures how the phase angle of the speech envelope synchronizes with the phase angle of the EEG. A significant strength of PLV is its direct involvement with the stimuli, allowing for investigation at the trial level rather than solely relying on frequency representation in the EEG. However, PLV is more biased by sample size compared to other measures of phase synchrony and has previously been outperformed by other connectivity measures, such as pairwise phase consistency [30, 49, 50]. We chose PLV for its computational efficiency and suitability for short stimuli, making it a pragmatic choice for our study design. There are robust methods to evaluate the neurophysiological processing of natural speech, such as the multivariate temporal response function (mTRF) framework [51]. This method has proven useful in elucidating processing differences at different acoustic and linguistic scales in older age [40], post-stroke aphasia [52], and cognitive decline [21]. However, we did not deem the mTRF method appropriate for our study because the short stimuli, split into many trials,

would have significantly limited both the performance of the algorithms and the cross-validation procedures typically used in the TRF domain.

**Speech tracking in different linguistic timescales led to narrow filtering.** Furthermore, we adapted the approach of using stimuli-specific linguistic timescales, as previously done by Keitel et al. [33]. While the use of stimuli-specific timescales is a strength of this study, it required very narrow filtering of the signals. Narrow filtering can lead to potential distortion of the signal and loss of important information, as well as reduced sensitivity to certain aspects of the data. This could result in the overestimation or underestimation of the effects of cognitive decline and hearing loss on speech tracking at different linguistic timescales.

## Conclusion

In this study, we investigated how cognitive decline and hearing loss interact to affect the neural processing of speech, with a particular focus on the role of contextual cues. Our findings indicate that cognitive decline is associated cognitive decline was associated with decreased response accuracy in a behavioral speech recognition task, aligning with our hypothesis. Neurophysiological responses showed more complex patterns, with cognitive decline and hearing loss impacting different components of speech processing. Specifically, we found that cognitive decline affected the P2 component in evoked potentials, while hearing ability modulated speech tracking strength at the word and phoneme rates. These results suggest that the neural mechanisms underlying speech processing are differentially affected by cognitive decline and hearing loss, with contextual cues playing a significant role in facilitating speech tracking at the syllable rate. Our study highlights the importance of considering both cognitive and auditory factors when examining speech processing in older adults. Furthermore, in our ageing society, a growing number of people will be affected by multimorbidity, and our results suggest that the interactions between common diseases should be further investigated.

Future research should aim to build on these findings by incorporating participants with distinct neurocognitive profiles. Such studies could use more comprehensive cognitive testing to corroborate findings suggested by the MoCA. Given the promising results of our study, further investigation into the interaction between cognitive decline, hearing loss, and contextual cues in speech processing is warranted. Understanding these interactions could lead to the development of more effective diagnostic tools and intervention strategies to support older adults with cognitive decline and hearing loss.

## Materials and methods

### Participants

We recruited 45 participants aged 60 years and older via an advertisement on Pro Senectute (prosenectute.ch). Please note that these are the same participants who took part in another experiment as part of the same study project (see [21, 22]). All participants were native Swiss German speakers, monolingual until at least the age of seven, right-handed, and retired. We conducted a preliminary telephone interview followed by a detailed on-site interview to select participants. We confirmed right-handedness using the Annett Hand Preference Questionnaire [53]. The inclusion criteria ensured that participants had no professional music careers, dyslexia, significant neurological impairments such as dementia or depression, severe or asymmetrical hearing loss, or use of hearing aids. The Ethics Committee of the Canton of Zurich approved this study (BASEC No. 2019-01400). All sessions were conducted at the Linguistic Research Infrastructure (LiRI, liri.uzh.ch) of the University of Zurich. Participant recruitment took place between November 23, 2022, and March 15, 2023. All participants provided written informed consent and received monetary compensation for their participation.

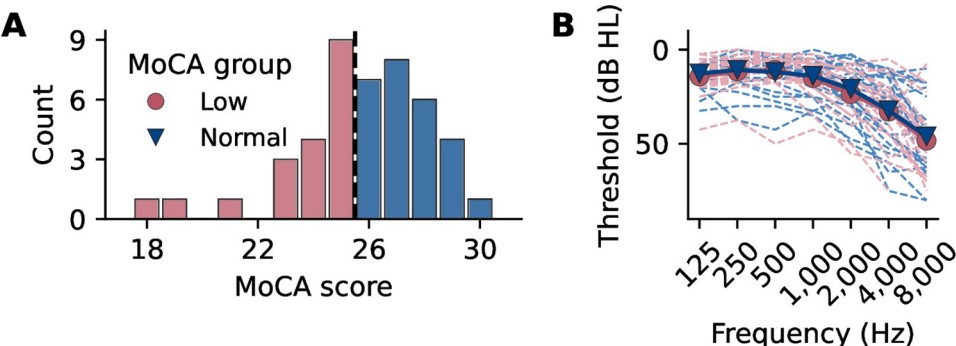

**Fig 4. MoCA scores distribution and audiometry results for participants. A** Histogram of participants' Montreal Cognitive Assessment (MoCA) scores, color-coded according to the MoCA groups resulting from this classification. The dashed line indicates the cut-off value of 26 points. **B** Individual hearing thresholds for each frequency, averaged by ear and color coded by MoCA group. The PTA values calculated from the audiogram did not differ between MoCA groups.

**Cognitive decline assessed through MoCA.** We grouped participants based on their cognitive performance using MoCA [31]. MoCA scores range from 0 to 30 points, with higher scores indicating better cognitive function. Using a cutoff score of 26 points for MCI [31], we divided participants into two groups: those with signs of cognitive decline (low MoCA group, 19 participants with 23.6±2.1 points, 12 women) and those with normal cognitive performance (normal MoCA group, 26 participants with 27.4±1.2 points, 14 women). The low MoCA group had an average age of 71.7±6.3 years (range: 60–82), and the normal MoCA group had an average age of 69±5.4 years (range: 60–83). The age difference between the groups was not statistically significant (two-tailed $t$-test: $t(43) = 1.6$, $p = 0.121$, $d = 0.48$). The distribution of MoCA scores is displayed in Fig 4A.

## Audiometry

We conducted audiometric testing using the Affinity Compact audiometer (Interacoustics, Middelfart, Denmark) with a DD450 circumaural headset (Radioear, New Eagle, PA, USA). Participants sat in a soundproof booth to minimize external interference. We measured pure-tone hearing thresholds for both ears at frequencies of 125, 250, 500, 1, 000, 2, 000, 4, 000, and 8, 000 Hz. We assessed overall hearing ability based on the four-frequency pure-tone average (PTA), calculated as the average of thresholds at 500, 1, 000, 2, 000, and 4, 000 Hz [32]. The average PTA across both ears was 20.0±11.6 dB HL in the low MoCA group and 19.6±11.6 dB HL in the normal MoCA group. Most participants had no to mild hearing impairment, though seven had moderate impairment, defined as a PTA exceeding 34 dB HL [36], with two in the low MoCA group and five in the normal MoCA group. No significant difference in hearing ability was observed between the groups (two-tailed $t$-test: $t(43) = 0.1$, $p = 0.815$, $d = 0.07$), but PTA correlated with age (Pearson's $r = 0.50$, $p < 0.001$, Fisher's $z = 0.55$). Individual hearing thresholds are shown in Fig 4B.

## Speech stimuli creation

In this study, participants listened to two sets of 60 Standard German matrix sentences, designed to sound like natural continuous speech and structurally inspired by the stimulus material reported by Keitel et al. [33]. We created sentences that either provided a supporting context, similar to the high-context sentences of the SPIN-R [26, 47, 48], or consisted of

randomly combined elements without context. Each sentence was at least six seconds long to ensure reliable speech tracking values.

Initially, we developed two sets of 160 matrix-style sentences: a context set and a random set. Each sentence followed a fixed grammatical structure with an initial clause and a relative clause constructed from four phrases. An example context sentence is: *Das Mädchen durfte zuschauen, wie die Fischerin die grossen Forellen mit einer langen Angel gefangen hat.* [The girl could watch how the fisherwoman caught the big trout with a long fishing rod]. Each matrix sentence included:

**Initial clause**: *Das Mädchen durfte zuschauen,*—The main clause, ending at the comma.

**Relative clause**: *wie die Fischerin die grossen Forellen mit einer langen Angel gefangen hat.*— This subordinate clause provides additional information about the main clause. It includes four phrases:

**Subject phrase**: *wie die Fischerin*—The subject of the action in the relative clause.

**Object phrase**: *die grossen Forellen*—The object of the action.

**Prepositional phrase**: *mit einer langen Angel*—Describes how or where the action took place.

**Verb phrase**: *gefangen hat*—Contains the past participle of the verb and the auxiliary verb.

Although the initial clause varied, from the subject phrase onward, all sentences consisted of exactly eleven words. Each sentence had at least 28 and at most 31 syllables and was semantically meaningful and easy to comprehend.

For the context set, sentences provided contextual congruity. The nouns in the relative and subject phrases hinted at the target noun in the prepositional phrase. For example, *Fischerin* [fisherwoman] and *Forellen* [trout] suggest *Angel* [fishing rod] as the prepositional phrase. To create the random set, we shuffled elements of the context sentences, ensuring grammatical correctness. For instance, a random sentence equivalent is: *Peter hat mir letzthin erzählt, dass der Wanderer den seltenen Tiger mit einer langen Angel gefangen hat.* [Peter recently told me that the hiker caught the rare tiger with a long fishing rod].

A professional native Swiss German-speaking actress voiced the sentences. Her voice had a mean fundamental frequency ($F_0$) of 210±8.3 Hz, and she read the sentences neutrally, maintaining consistent rhythm and intonation. Recordings took place at the video studios of the University of Zurich (zi.uzh.ch/media/video-and-multimedia), digitized at a sampling rate of 48 kHz. We manually edited the recordings to remove background noise, breathing sounds, and other artifacts, and normalized the recordings to 70 dB root mean square (RMS) amplitude.

To validate the context sentences, we presented them to 19 volunteers, asking them to fill in the masked last noun in the prepositional phrase. We selected the recordings of the 62 highest-rated sentences, where volunteers correctly identified the target noun at least 80% of the time (average 93.5±0.1%). We supplemented these with 62 sentences from the random set. Our final stimulus set consisted of 124 sentences, with participants listening to 120 sentences (60 context, 60 random) during the main experiment, and four additional sentences (two from each set) for practice trials.

## Stimulus presentation and speech recognition task

We used Presentation®software (Neurobehavioral Systems, Inc., Berkeley, CA, USA) to display instructions and play auditory stimuli. Participants listened to the sentences bilaterally

through ER3 insert earphones (Etymotic Research, Inc., Elk Grove Village, IL, USA), with the sound level calibrated to 70 dB peSPL. To avoid systematic effects of sentence length or content, we presented the matrix sentences in a fully randomized order. To keep participants engaged and to monitor their attention, we included a simple behavioral speech recognition task. After each sentence, participants performed a forced-choice task, identifying if one of four words was present in the sentence. To prevent memorization of word locations within the fixed sentence structure, we varied the word location, asking about the noun of the subject phrase, the verb of the object phrase, or the noun of the prepositional phrase. These target locations were counterbalanced across the 120 sentences, and the response key was systematically co-randomized during presentation. Participants indicated their responses by pressing the corresponding key marked with a sticker on the keyboard.

The sentences lasted on average 6.8±0.4 seconds. Before the experiment, we conducted a practice session with four trials to ensure participants understood the task. The sentences from the practice trials were not included in the analyses. Randomization of sentences began after the training session. The trials were presented in three blocks of 40 sentences each, with a short break between blocks. These two breaks lasted approximately five minutes each and were used to maintain participants' concentration.

### Neurophysiological recording

During the listening task, we recorded the participants' brain activity using the Biosemi ActiveTwo system (Biosemi, Amsterdam, The Netherlands) with 32 electrodes (10–20 system), four external electrodes, and a sampling rate of 16.384 kHz. The high sampling rate was due to the longer experiment, which also included subcortical measures discussed in a separate paper [21], but left the data for the current study unaffected. We placed two external electrodes above and below the right eye to measure the electrooculogram (EOG) and two on the left and right mastoids. Throughout the experiment, we monitored and maintained electrode offsets below 20 $\mu$V. Recordings were conducted in a soundproof, electromagnetically shielded booth. We instructed participants to focus on a fixation cross displayed on a screen and to minimize movement, especially when the cross was visible.

### EEG preprocessing

We performed all EEG and speech file processing in Python 3.11.9 using MNE-Python [54]. First, we visually inspected the raw EEG data for noisy electrode channels. On average, we removed 1.2±1.8 channels per participant and then referenced the electrode signals to the average of the external mastoid electrodes. To expedite computation given our high sampling rate, we downsampled early in the process. However, to preserve the precision of our wav-cued triggers, we segmented the EEG before downsampling. We had triggers for the onset of the speech and the target noun, which we used to segment the continuous EEG into epochs. All filters described here were infinite impulse response (IIR), specified as zero-phase non-causal 3$^{rd}$ order Butterworth filters, unless otherwise noted. We applied an anti-alias filter at 170.7 Hz (high cut at $\frac{1}{3}$ of the sampling rate) to the continuous EEG. To remove power line noise, we applied IIR notch filters at 50, 100, and 150 Hz (bandstop filter, 2$^{rd}$ order) with 5 Hz notch widths. Next, we segmented the continuous EEG into two epoch sets: epochs locked to speech onset ("onset epochs") and epochs locked to target noun onset ("target epochs"). Onset epochs spanned from −2 to 10 s, and target epochs from −1 to 5 s. We chose longer segmentation windows to reduce filtering artifacts during subsequent processing and to provide more data for the Independent Component Analysis (ICA) decomposition. We then decimated the epochs to 512 Hz.

To remove artifacts, we applied ICA. For this process, we created a copy of the epochs and high-pass filtered the data at 1 Hz, which facilitates the ICA decomposition [55]. We used the Picard algorithm [56] to perform ICA on the epoch copy with 1,000 iterations, aiming to capture 99.9% of the signal variance. We further enhanced ICA performance with five iterations of the FastICA algorithm [57]. After ICA fitting, we automatically labeled components associated with eye-related artifacts using the external EOG electrodes as a reference. We manually labeled components associated with muscle activity or singular artifacts based on topography, temporal occurrence, and frequency spectrum. We zeroed out the identified components in the original epochs and subsequently interpolated bad channels. On average, we excluded 2.4 ±0.8 components per participant. Cleaned epochs were cached for further analysis.

## Evoked potential analyses

We conducted evoked potential analysis to assess neural responses to speech onset and target noun onset. Specifically, our goal was to identify AEP-related P1, N1, and P2 components condition-related N400 components. To this end, we filtered the cleaned onset and target epochs with a bandpass filter from 0.1 to 30 Hz. We removed all trials to which participants responded incorrectly and applied a baseline correction using the −200 to 0 ms interval before speech and target noun onset. Finally, we cropped the epochs to −200 to 1,000 ms relative to speech and target noun onset, respectively.

**P1, N1, and P2 components.** To obtain the AEPs, we averaged the epochs across all trials. We focused on an auditory-relevant frontotemporal cluster consisting of six electrodes: F3, FC1, FC5, FC6, FC2, and F4, consistent with previous studies [22, 58, 59]. We analyzed the P1, N1, and P2 components, which are typically observed in the AEP. For each participant, we extracted the mean amplitude and latency of each component from the grand average waveform. Specifically, we defined the following time windows for each component: 25 to 100 ms for P1, 50 to 150 ms for N1, and 150 to 300 ms for P2 [60]. Within these windows, we identified the positive and negative peaks using the maximum and minimum values, respectively. Subsequently, for each peak, we extracted the latency and the mean amplitude within a 50 ms window centered on the peak.

**N400 component.** To extract the N400 component, we averaged epochs separately for context and random sentences. We focused on a centroparietal cluster of six electrodes: Cz, CP1, CP2, Pz, and P3. This electrode selection aligns with previous studies showing that the N400 component is most prominent at centroparietal sites [24, 61, 62]. The N400 typically manifests as a sustained negative deflection occurring between 300 and 600 ms post-stimulus [61]. For each participant, we identified the most negative amplitude within this window in the random condition and extracted both the latency of this deflection and the mean amplitude within a 100 ms window centered on this point. We then used the same latency to extract the mean amplitude in the context condition. This approach ensures precise measurement of the N400, capturing its variability in both latency and amplitude across conditions [60].

## Speech tracking analyses

The goal of this analysis is to investigate speech tracking at linguistic timescales inherent in the speech stimulus material, rather than using generic frequency bands like the $\delta$ or $\theta$ bands. We adopted this approach from Keitel et al. [33], who demonstrated that using stimulus-specific timescales allows for a more precise alignment of neural activity with the temporal characteristics of speech. This method enhances the interpretation of neural speech tracking by focusing on the actual linguistic properties of the speech material, leading to more accurate insights into the neural mechanisms of speech processing.

**Determining linguistic timescales.** For this analysis, we focused on the timescales of phrases, words, syllables, and phonemes present in the speech material [33]. We determined the phrase rate by dividing the number of phrases—always five, including the initial clause and the four phrases in the relative clause—by the duration of the sentence. Similarly, the syllable rate, which we controlled strictly during stimulus creation, was calculated as the number of syllables divided by the sentence duration. To obtain the word and phoneme rates, we used the Montreal Forced Aligner [63] and its German pronunciation dictionary [64] to align the speech recordings with the corresponding orthographic transcriptions. The MFA output provided the timing of word and phoneme boundaries for each segment, which we used to calculate the word and phoneme rates by dividing the number of word or phoneme boundaries by the sentence duration.

The average phrase rate was 0.73 Hz (range: 0.62–0.85 Hz). The average word rate was 2.40 Hz (range: 1.97–2.78 Hz). The average syllable rate was 4.34 Hz (range: 3.46–5.01 Hz). The average phoneme rate was 10.77 Hz (range: 8.95–12.24 Hz).

**Speech envelope extraction.** To analyze speech tracking, we extracted the envelope from each stimulus speech signal. Following the procedure outlined by Oderbolz et al. [50], we applied a Gammatone filterbank [65] with eight subbands to the speech stimuli, with center frequencies ranging from 20 Hz to 20 kHz, spaced equally on the equivalent rectangular bandwidth scale. The output was then half-wave rectified and compressed by a factor of 0.6. We averaged the subbands to obtain the envelope. Finally, we resampled the envelope from 48 kHz to 512 Hz after applying an anti-aliasing filter at 170.7 Hz (3$^{rd}$ order) and stored the envelopes for further analysis.

The modulation spectrum shown in Fig 1A reflects the temporal modulation of the speech signals (across all 120 sentences), calculated using the procedure described by Ding et al. [50, 66]. Specifically, we computed the Discrete Fourier Transform (DFT) of the envelope for each speech stimulus using the Fast Fourier Transform (FFT). The modulation spectrum is the RMS of the DFT across all envelopes.

**Timescale-specific speech tracking.** We used the PLV to quantify the alignment between neural activity and the speech envelope at each linguistic timescale. The PLV is a measure of phase consistency across trials, ranging from 0 (no phase consistency) to 1 (perfect phase consistency) [49]. The formula for the PLV is given by

$$\text{PLV}(t) = \left| \frac{1}{N} \sum_{n=1}^{N} e^{j\Delta\phi(t,n)} \right|, \tag{1}$$

where $N$ is the number of trials, $\Delta\phi(t, n)$ is the phase difference between the speech envelope and the EEG signal at time $t$ and trial $n$, and $j$ is the imaginary unit.

To analyze speech tracking for each of the four rates, we filtered the onset epochs with a bandpass filter corresponding to each timescale: 0.62–0.85 Hz for phrase rate, 1.97–2.78 Hz for word rate, 3.46–5.01 Hz for syllable rate, and 8.95–12.24 Hz for phoneme rate. Again, we removed all trials with incorrect responses and applied a baseline correction using the −200 to 0 ms interval before speech onset. To ensure consistency, both the speech and EEG signals were cut or padded to match timepoints 0 (onset) to the mean stimulus length of 6.8 s. To speed up computation, we downsampled the epochs and envelopes to 128 Hz. For each envelope and EEG signal, we computed the Hilbert transform and extracted the instantaneous phase as the phase angle of the resulting analytic signal. We then calculated the PLV between the speech envelope and the EEG phase at each linguistic timescale and electrode.

**Electrode clusters.** Following an approach we adopted in a previous study [22], we preselected nine midline electrodes for detailed analysis. These electrodes—F3, Fz, F4, C3, Cz, C4,

P3, Pz, and P4—were grouped into three regional clusters: frontal (F), central (C), and parietal (P). This method enabled us to incorporate the activity of individual electrodes in our analysis while also conducting hierarchical modeling at the cluster level. The chosen electrodes were strategically positioned near the auditory cortex, targeting their relevance to neural speech processing. By covering a broad scalp area, we avoided making prior assumptions about the exact location of neural generators.

## Statistical analyses

All statistical analyses reported in the Materials and Methods and Results sections were conducted in R (version 4.4.0). For group and condition comparisons, we used paired and unpaired Student's $t$-tests, reporting $t$-values with their degrees of freedom, $p$-values, and effect sizes as Cohen's $d$.

For linear modeling, both fixed effects-only and mixed effects models, we normalized all continuous predictor variables, specifically PTA and age, using $z$-transformations. Factors were coded using treatment contrasts. MoCA group was treated as a categorical variable with two levels: *normal* and *low*, with *normal* as the reference level. Condition was a categorical variable with levels *context* and *random*, with *context* as the reference level. The cluster variable included in the speech tracking analysis was coded as a categorical variable with three levels: *F* (frontal), *C* (central), and *P* (parietal), with *F* as the reference level.

We implemented the generalized linear mixed models (GLMMs) and linear mixed models (LMMs) using the `lme4` package [67] and estimated $p$-values using the `lmerTest` package [68]. To obtain confidence intervals for the fixed effects, we bootstrapped the data using the Wald method with 5, 000 simulations.

**Behavioral analyses.** We used a GLMM to examine participants' performance on the reponse task after each sentence. The model accounted for the nested data structure, with stimuli nested within participants, using the following formula:

$$\text{Correct} \sim \text{MoCA group} \times \text{PTA} + \text{condition} + \text{age} +$$

$$(1 \mid \text{stimulus ID}) + (1 \mid \text{participant ID}). \tag{2}$$

Correct represented the binary dependent variable, coded as 0 for incorrect responses and 1 otherwise. Fixed effects included MoCA group, PTA, condition, and age, as well as interactions between MoCA group and PTA. Random effects accounted for variability at the participant and stimulus levels. We reported odds ratios (ORs), their 95% confidence intervals, and $p$-values for the fixed effects.

**Evoked potential analyses.** To investigate if the amplitudes of P1, N1, and P2 components, as well as the difference in N400 negativity, differed between the low and normal MoCA groups, we conducted a linear regression analysis. The linear model was configured with the following formula:

$$\text{Amplitude} \sim \text{MoCA group} \times \text{PTA} + \text{age}, \tag{3}$$

where the amplitude represented the mean amplitude in microvolts ($\mu$V) of the respective component or the N400 difference. We calculated the N400 difference as the difference in the mean amplitude within the 100 ms window centered around the N400 latency in the context response minus the mean amplitude of the N400 in the random condition. Fixed effects in the model included MoCA group, PTA, and age, as well as an interaction term between MoCA group and PTA. We included age as a covariate in the model to account for age-related effects on the AEPs and N400. This model was run separately for each component and the N400

difference. We reported the estimated coefficients, their 95% confidence intervals, and $p$-values for the fixed effects.

**Speech tracking analyses.** To investigate the PLV at each linguistic timescale, we conducted a LMM analysis. The structure of the LMM accounts for the nested data structure, with electrodes and stimuli nested within participants. The LMM was configured with the following formula:

$$\text{PLV} \sim \text{MoCA group} \times \text{PTA} \times \text{condition} + \text{cluster} + \text{age} +$$

$$(1 \mid \text{stimulus ID}) + (1 \mid \text{participant ID}). \tag{4}$$

PLV represented the phase-locking value at each linguistic timescale. Fixed effects in the model included MoCA group, PTA, condition, cluster, and age. Interactions between MoCA group, PTA, and condition were included to explore their combined effects on PLV. Again, we included random effects to account for variability at the participant and stimulus levels. A random intercept was specified for each participant to capture individual differences in baseline PLV. A random intercept was included for each stimulus to account for variability due to the specific speech stimuli used in the experiment. For each model, we reported the estimated coefficients, the 95% confidence intervals, and the $p$-values for the fixed effects. Confidence intervals were bootstrapped using the Wald method with 5, 000 simulations.

Post-hoc pairwise comparisons were assessed using the EMM method through the `emmeans` package in R. EMMs provide estimates of the marginal means for each factor level, adjusted for the other factors in the model. For interactions, we examined the simple effects within each level of the interacting factors. To understand the interaction between MoCA group and condition, we compared the EMMs of the MoCA groups within each condition separately. Similarly, to investigate the interaction between PTA and condition, we fixed PTA at three standardized levels ($z = -1$, 0, and 1) and compared the EMMs across these levels within each condition. The pairwise comparisons were adjusted for multiple testing using the Tukey method, which controls the family-wise error rate. We reported the EMMs and $p$-values for these comparisons.

## Acknowledgments

We thank Elainne Vibal for her help with data collection and Andrew Clark from LiRI for his support in setting up the EEG experiment. We are grateful to Chantal Oderbolz for her inputs to the PLV-based speech tracking analyses. Special thanks to Kathrin Frei for voicing the stimulus material and Vanessa Frei for her assistance with the studio recordings. We also appreciate Jens Hjortkjær's valuable input on this work, provided at the International Conference on Cognitive Hearing Science for Communication (CHSCOM 2024) in Linköping. Finally, we thank Jill Kries and the other anonymous reviewers for their constructive feedback on the manuscript, which significantly improved the quality of the paper.

## Author Contributions

**Conceptualization:** Elena Bolt, Katarina Kliestenec, Nathalie Giroud.

**Data curation:** Elena Bolt, Katarina Kliestenec.

**Formal analysis:** Elena Bolt.

**Funding acquisition:** Nathalie Giroud.

**Investigation:** Elena Bolt, Katarina Kliestenec.

**Methodology:** Elena Bolt, Katarina Kliestenec.

**Project administration:** Elena Bolt.

**Resources:** Elena Bolt.

**Software:** Elena Bolt.

**Supervision:** Nathalie Giroud.

**Validation:** Elena Bolt.

**Visualization:** Elena Bolt.

**Writing – original draft:** Elena Bolt.

**Writing – review & editing:** Elena Bolt, Katarina Kliestenec, Nathalie Giroud.

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
