## [Decision Letter · Decision Letter 0]

30 Sep 2024

PONE-D-24-30989Hearing and cognitive decline in aging differentially impact neural tracking of context-supported versus random speech across linguistic timescalesPLOS ONE

Dear Dr. Bolt,

Thank you for submitting your manuscript to PLOS ONE. After careful consideration, we feel that it has merit but does not fully meet PLOS ONE’s publication criteria as it currently stands. Therefore, we invite you to submit a revised version of the manuscript that addresses the points raised during the review process. Both reviewers and myself are very impressed with your submission. The manuscript is well written, thorough, and covers an interesting and timely topic. 

We look forward to receiving your revised manuscript.

Kind regards,

Kathrin Rothermich

Academic Editor

PLOS ONE

Journal Requirements:

plos.org/plosone/s/file%3fid=wjVg/PLOSOne_formatting_sample_main_body.pdf%20andWhen submitting your revision, we need you to address these additional requirements.

3. Thank you for stating the following financial disclosure: “This study was funded by the Swiss National Science Foundation, SNSF, www.snf.ch, grant number PR00P1 185715 to Nathalie Giroud”

4. We note that your Data Availability Statement is currently as follows: “All relevant data are within the manuscript and in Supporting Information files.”

Please confirm at this time whether or not your submission contains all raw data required to replicate the results of your study. Authors must share the “minimal data set” for their submission. PLOS defines the minimal data set to consist of the data required to replicate all study findings reported in the article, as well as related metadata and methods (https://journals.plos.org/plosone/s/data-availability#loc-minimal-data-set-definition). For example, authors should submit the following data: - The values behind the means, standard deviations and other measures reported; - The values used to build graphs; - The points extracted from images for analysis. Authors do not need to submit their entire data set if only a portion of the data was used in the reported study. If your submission does not contain these data, please either upload them as Supporting Information files or deposit them to a stable, public repository and provide us with the relevant URLs, DOIs, or accession numbers. For a list of recommended repositories, please see https://journals.plos.org/plosone/s/recommended-repositories. If there are ethical or legal restrictions on sharing a de-identified data set, please explain them in detail (e.g., data contain potentially sensitive information, data are owned by a third-party organization, etc.) and who has imposed them (e.g., an ethics committee). Please also provide contact information for a data access committee, ethics committee, or other institutional body to which data requests may be sent. If data are owned by a third party, please indicate how others may request data access.

Reviewers' comments:

Reviewer's Responses to Questions

**Comments to the Author**

1. Is the manuscript technically sound, and do the data support the conclusions?

Reviewer #1: Yes

Reviewer #2: Yes

2. Has the statistical analysis been performed appropriately and rigorously? 

Reviewer #1: Yes

Reviewer #2: Yes

3. Have the authors made all data underlying the findings in their manuscript fully available?

Reviewer #1: Yes

Reviewer #2: Yes

4. Is the manuscript presented in an intelligible fashion and written in standard English?

Reviewer #1: Yes

Reviewer #2: Yes

5. Review Comments to the Author

Reviewer #1: Thank you for the opportunity to review this well-written manuscript on a fascinating topic.

I only have a few minor comments for the authors:

The manuscript contains question marks instead of key references, which must be addressed and verified before publication.

The author should aim for better figure and table placement in the final manuscript, as the placement in this version created some confusion. This said the results were generally well presented and well summarised.

Any minor technical or conceptual issue I had about the protocol was addressed correctly in the limitations section.

Reviewer #2: An interesting and well-written paper. Most caveats have already been identified in the discussion. Here are my comments:

- line 47: reference missing

- line 51-53: “our previous study” comes as a surprise to the reader, reformulate

- line 53: reference missing

- line 85: “discrete chunks of acoustic units”, shouldn’t it rather be linguistic units?

- line 88: add number of participants

- line 89: explain what matrix sentences are, not everyone knows that

- line 134-136: change order of sentences: “The main goal of this behavioral speech recognition task after each stimulus was to maintain their attention and ensure that they listened. Participants performed well, answering correctly in 96 ± 4% of the trials. No participant scored below 85% correct responses. This task was designed to be easy, and we excluded all trials with incorrect responses from neurophysiological analyses.”

- line 160: can you add a plot to the supplementary information with the AEPs of each subject individually laid over each other (averaged across the relevant EEG channels)

- line 209-211: can you elaborate why you assume that why the parietal cluster result would originate from frontocentral electrodes? And the formulation is a bit cryptic, i.e., “Across the scalp” to me means all electrodes, but then you talk about the parietal cluster. Also I think the idea of also including clusters in your statistical analysis is not explained properly before you report the results, so please do that.

- line 225-227: idem previous comment

- Figure 3B: add individual data points please, you can make them transparent if you think the plot becomes too busy

- line 230-232: are the post hoc comparisons not significant (even though the interaction is) due to correction for multiple comparisons? I am just trying to understand how come none of the post hoc comparisons was significant.

- line 259-261: “As hearing ability increased, tracking at the phoneme rate increased.” this is the opposite of what the plot 3B shows, isn’t it? You measure PTA in dbHL, that means that the higher the PTA value is, the worse their hearing ability is. Thus, it should be “As hearing ability decreased, tracking at the phoneme rate increased.” Make sure throughout the paper that you get this correct.

- line 267-268: “a lower chance of correct responses”, this seems to be a bit of a complicated formulation to say that they ha decreased response accuracy. Can you change that throughout the manuscript please?

- line 290-293: maybe add already here that the absence of an effect of cognitive decline on N400 might be related to your low MoCA group being very mild, with 9 participants actually scoring only one point below the clinical cut-off threshold

- line 341-344 and line 361-364: these speculations about non-significant effects are a bit too speculative, I would remove that

- section Strengths and limitations of the study: add that you have a limited variability of hearing loss in your sample, which might veil effects of PTA

- line 427: reference missing

- line 441-442: use “decreasing response accuracy” instead of “lower chance of correct responses”

- line 465: reference missing

- line 580: reference missing

- line 710-711: “ensuring” is a bit too strong, replace with “targeting”, or “trying to target”

6. PLOS authors have the option to publish the peer review history of their article (what does this mean?). If published, this will include your full peer review and any attached files.

Reviewer #1: No

Reviewer #2: **Yes: **Jill Kries, PhD

---

## [Author Response · Author response to Decision Letter 0]

28 Oct 2024

The responses to the reviewers are documented in the letter.

---

## [Editor Report · Decision Letter 1]

1 Nov 2024

Hearing and cognitive decline in aging differentially impact neural tracking of context-supported versus random speech across linguistic timescales

PONE-D-24-30989R1

Dear Dr. Bolt,

We’re pleased to inform you that your manuscript has been judged scientifically suitable for publication and will be formally accepted for publication once it meets all outstanding technical requirements.

Kind regards,

Kathrin Rothermich

Academic Editor

PLOS ONE
---

## [Editor Report · Acceptance letter]

26 Nov 2024

PONE-D-24-30989R1 

PLOS ONE

Dear Dr. Bolt, 

I'm pleased to inform you that your manuscript has been deemed suitable for publication in PLOS ONE. Congratulations! Your manuscript is now being handed over to our production team.

Kind regards, 

on behalf of

Dr. Kathrin Rothermich 

Academic Editor

PLOS ONE